# Mood symptoms predict COVID-19 pandemic distress but not vice versa: An 18-month longitudinal study

**Benjamin A. Katz** [ID]*, **Iftah Yovel** [ID]

The Hebrew University of Jerusalem, Jerusalem, Israel

* Benjamin.katz@mail.huji.ac.il

## Abstract

The COVID-19 pandemic has had medical, economic and behavioral implications on a global scale, with research emerging to indicate that it negatively impacted the population's mental health as well. The current study utilizes longitudinal data to assess whether the pandemic led to an increase in depression and anxiety across participants or whether a diathesis-stress model would be more appropriate. An international group of 218 participants completed measures of depression, anxiety, rumination and distress intolerance at two baselines six months apart as well as during the onset of the COVID-19 pandemic exactly 12 months later. Contrary to expectations, depression, rumination, and distress intolerance were at equivalent levels during the pandemic as they were at baseline. Anxiety was reduced by a trivial degree ($d = .10$). Furthermore, a comparison of quantitative explanatory models indicated that symptom severity and pandemic-related environmental stressors predicted pandemic-related distress. Pandemic-related distress did not predict symptom severity. These findings underscore the necessity of longitudinal designs and diathesis-stress models in the study of mental health during the COVID-19 pandemic. They also emphasize that individuals with higher rates of baseline psychopathology are as particularly at risk for higher levels of distress in response to disaster-related stressors.

**Data Availability Statement:** All data, and analysis syntax are available from https://osf.io/sjp4a/.

**Funding:** This research was supported by the Israeli Science Foundation Grant 886/18 to IY. (https://www.isf.org.il). The funders had no role in

## Introduction

COVID-19 (SARS-CoV-2) is a novel, disproportionately infectious and lethal strain of coronavirus that has become a global pandemic over the course of early 2020 [1]. Theoretical articles have highlighted how the COVID-19 pandemic includes stressors uniquely fit to negatively impact mental health on a population level as well [2–6]. The high levels of health and financial uncertainly salient to the pandemic are strongly linked to stress [7], internalizing pathology [8] and externalizing pathology [9]. Furthermore, early restrictions on travel and gathering, self-isolation, and quarantine have led to periods of loss of social support and loneliness, both of which predict depression in particular [10, 11].

Researchers and public health specialists raised concerns that this time of acute stress would bring about a global spike in mental illness [12]. In doing so, they implicitly argue in

study design, data collection and analysis, decision to publish, or preparation of the manuscript.

**Competing interests:** The authors have declared that no competing interests exist.

favor of a general-stressor model, pointing out that the many disruptions and challenges presented by the COVID-19 pandemic have led to greater levels of cumulative stress across the population (Fig 1a). This population-level stress would transdiagnostically increase levels of mental illness on a population level [13]. Indeed, infection with COVID-19 has indeed been found to negatively impact mental health [14, 15] and pandemic-related stress has been found to be associated with elevated symptom severity [16, 17]. However, recent large-scale studies have found less support for the general stressor model on a population level, even at the beginning of the pandemic, a period of great stress and uncertainty [6, 18, 19]. Indeed, populations tend to be quite resilient in the face of disaster-related stress (for review, see [20]). For example, in 2012, Hurricane Sandy-related stress only predicted elevated symptoms among children predisposed to symptom-relevant affect; those high in temperamental sadness in a prior assessment showed elevated levels of depressive symptoms while those high in temperamental fearfulness showed elevated levels of anxiety symptoms [21]. Similarly, the COVID-19 pandemic may serve as a trigger for those with vulnerabilities to specific disorders as opposed to as a population-level stressor [22].

Furthermore, the theorized relationship between pandemic-related distress and psychopathology may follow an opposite causal direction as that presumed in a general-stressor model. Loneliness caused by self-isolation may lead to greater levels of depression as hypothesized [10, 23]. However, in a diathesis-stress model (See Fig 1b), the opposite causal direction is also possible [13]. In such a case, individuals with more severe depression are themselves more sensitive to distress [24]. Those with more severe baseline symptom severity may be more sensitive to the loneliness experienced during self-isolation, particularly at the beginning [18, 25]. Thus, in such a scenario, the observed association between depression and pandemic-related loneliness would still exist [16]. However, it would not be because pandemic-related loneliness led to greater levels of depression. Rather, in such a model, those who entered the pandemic with higher levels of depression would feel greater levels of loneliness during lockdown.

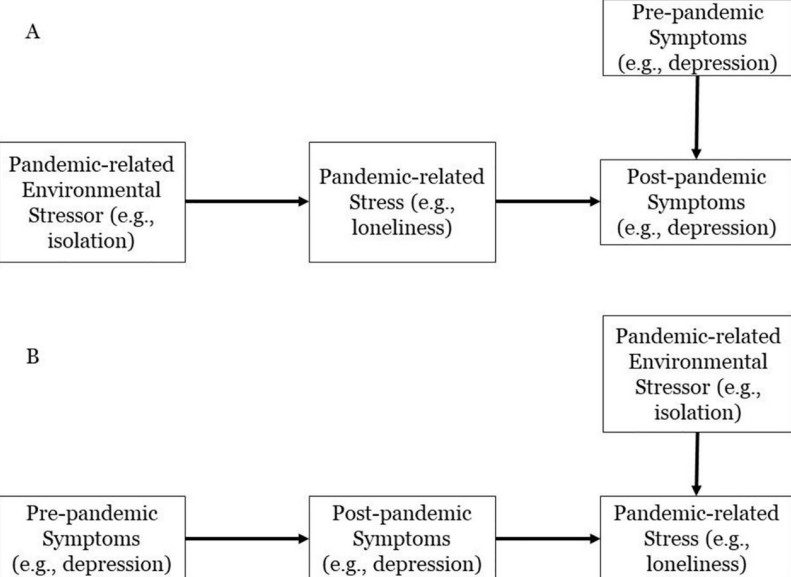

**Fig 1. Alternate explanations for the association between COVID-19 pandemic-related stress and mood disorders.** Fig 1a presents a general stressor model, where pandemic-related environmental stressors lead to stress, which in turn leads to a change in symptom severity. Fig 1b presents a diathesis-stress model, where pandemic-related stress is predicted by both current levels of symptom severity and pandemic-related environmental stressors.

Testing these alternative hypotheses demands certain research design prerequisites [26]. Stress-oriented models of psychopathology are inherently longitudinal [13] and should ideally include comparable baseline measures taken before the stressor was introduced [27]. Most disaster research, however, is either cross-sectional or longitudinal only following the disaster [28]. This is true for much research on the COVID-19 pandemic as well [14]. Furthermore, an assessment of psychiatric symptom severity must also statistically control for the confounding roles of environmental stressors and the distressed reactions to these stressors [13]. Thus, in order to assess the COVID-19 pandemic's negative impact on mental health, it is necessary to identify participants with pre-pandemic baseline data available, and to separately assess clinical symptoms, environmental stressors, and subjective distress. In doing so, the COVID-19 pandemic's role in mental health may be quantified, setting the groundwork for subsequent empirically-based interventions.

### Current study

The current study compared two opposing models of the COVID-19 pandemic's effect on depression and anxiety during a peak time of pandemic-related fatalities. To do so, we contacted participants who had participated in a previous six-month longitudinal study on emotion regulation, depression and anxiety and invited them one year following the final assessment to complete these measures again along with measures of stressors related to COVID-19 and the corollary public health interventions (e.g., self-isolation). The pandemic assessment period occurred between April 15, 2020 and April 20, 2020, during the 5-day period with the highest number of deaths per day in the United States in the first half of 2020 and the second-highest period in the United Kingdom (which immediately followed the highest one; see Fig 2 [29]). Thus, we were able to examine participants' levels of depression, anxiety, rumination and distress intolerance during the height of the COVID-19 pandemic, while also having two baseline measures from exactly one year and 18 months prior.

We had originally hypothesized a general stressor model, that (a) there would be no differences in the measures between the two baseline time-points, (b) there would be a group-level increase in levels of symptom severity and clinically relevant measures, and (c) increases in symptom severity would be predicted by participants' subjective distress related to COVID-19. However, after unexpectedly finding virtually no change in the measures between baseline and during the pandemic, we conducted a series of post-hoc analyses in order to confirm their equivalence across assessments.

Finally, we directly evaluated the general stressor model against the alternate, diathesis-stress model using two alternate explanatory models for how the COVID-19 pandemic relates to symptom severity beyond baseline. According to the first approach [10], pandemic-related environmental stressors would cause subjective distress, which in turn would predict symptom severity beyond baseline levels (Fig 3a and 3b). According to the second approach [20], symptom severity and environmental stressors would independently predict pandemic-related distress (Fig 4a and 4b). We quantified the likelihood of these approaches using two sets of path analyses that operationalized each approach and offered fit statistics for each model.

## Method

### Participants

The current sample consists of 218 participants (women = 118, men = 97, other/not applicable = 3) involved in an ongoing longitudinal study (see Procedure below). They represented a wide range of ages ($M$ = 42.87, $SD$ = 13.09, range = 19–75), with 102 participants in the United States, 100 in the United Kingdom, 15 in Canada, and one in Ireland. In order to ensure that

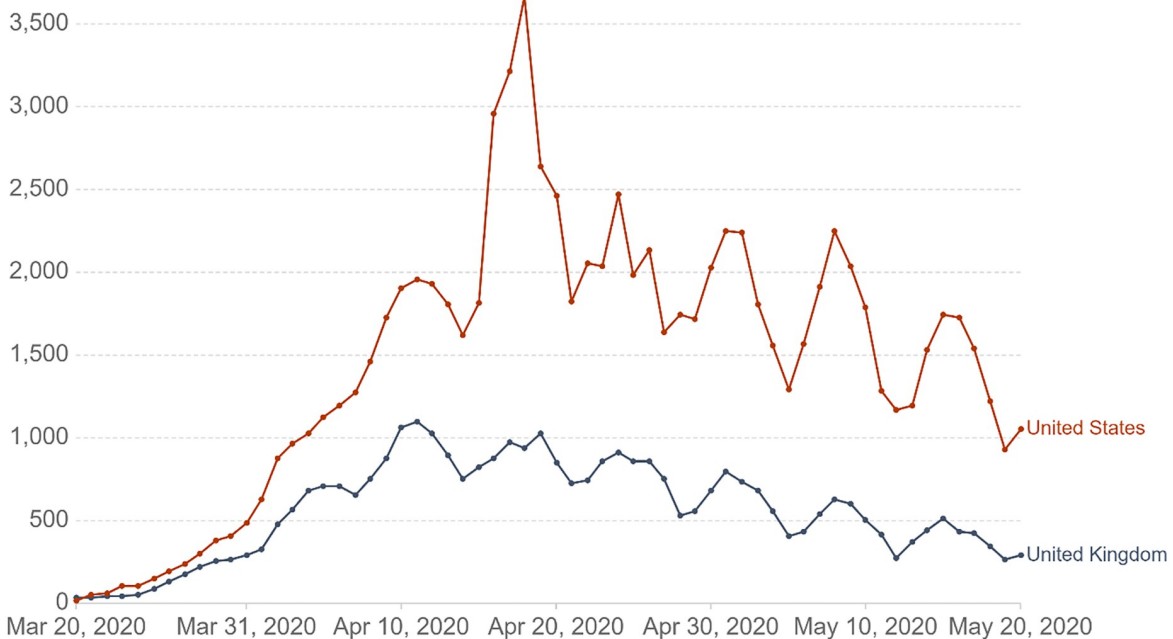

**Fig 2. Daily confirmed deaths of COVID-19 between March 20, 2020 and May 20, 2020.** Figure retrieved on September 26, 2021 from OurWorldInData.org/coronavirus, which visualized data reported by the European Center for Disease Control. The red rectangle indicates the dates in which the third assessment took place, between April 15 and April 20.

results were not biased by the Canadian and Irish participants, analyses were replicated for only participants from the United States and United Kingdom. Average scores on the measures and effect sizes remained the same as in the main analysis. See S1 File for the full output. Further demographic data is available in S1 Table.

## Materials

**Depression anxiety and stress scale-21, depression and anxiety subscales (DASS-21-D & DASS-21-A) [30, 31].** The DASS-21 is a widely used measure of affective symptoms experienced during the previous week and has been found to successfully track symptom change following natural disasters [32]. It consists of three seven-item subscales, measuring depression, anxiety, and stress. The first two subscales were included in the current study. The Depression subscale assesses of sadness and anhedonia (e.g., "I felt down-hearted and blue") and the Anxiety subscale assesses somatic experiencing of anxiety and fear (e.g., "I felt I was close to panic"). Each item was rated on a scale of 0 (= does not apply to me) to 3 (= applies to me very much, or most of the time). Subscale scores were calculated by summing together the items, with a possible range of 0–21, and higher scores indicating greater levels of depression or anxiety, respectively. In the current study, they showed very good-to-excellent reliability at all time points (Cronbach's alphas Anxiety = .86-.88; Depression = .94-.95).

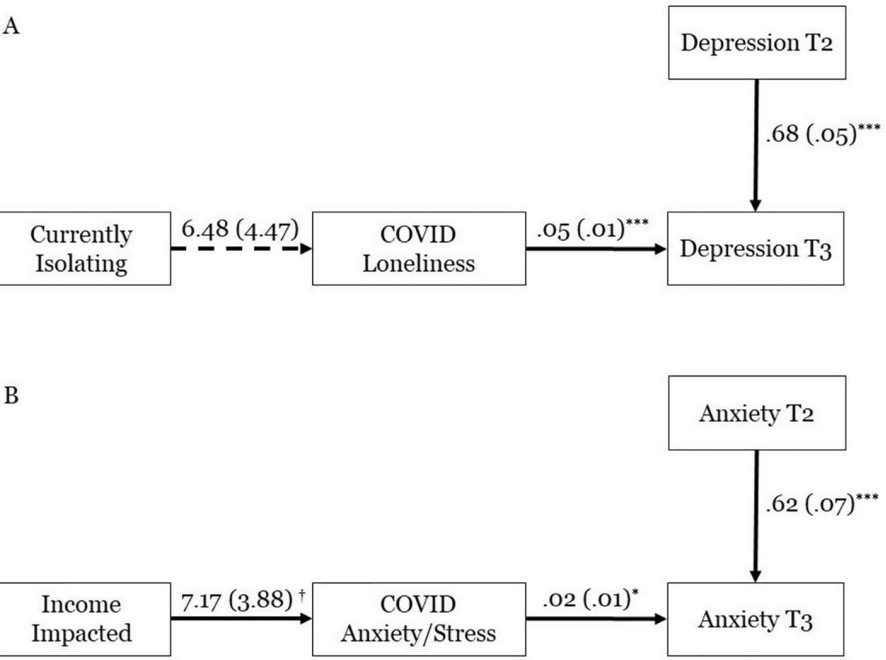

**Fig 3.** a and b. Structural equation model where COVID-19 loneliness/stress predicts depression/anxiety beyond baseline. *** $p < .001$, * $p \leq .05$, † $p < .10$. T2 –Data collection at Time 2, April 15–22, 2019. T3 –Data collection at Time 3, April 15–20, 2020. The above models were bad fits for the data (e.g., CFI = .000) and were therefore rejected.

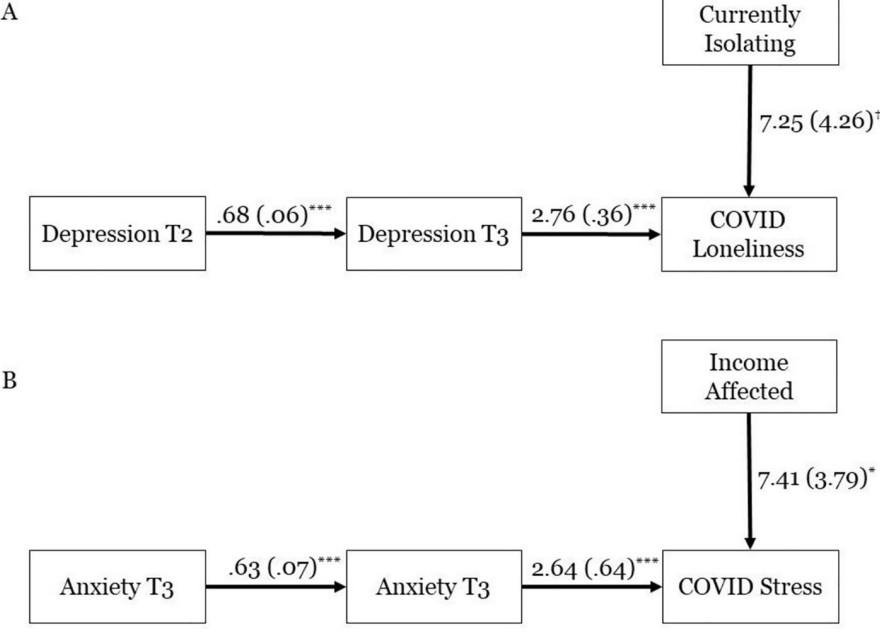

**Fig 4.** a and b. Structural equation model where COVID-19 depression/anxiety predicts COVID-19 loneliness/stress. *** $p < .001$, * $p \leq .05$, † $p < .10$. T2 –Data collection at Time 2, April 15–22, 2019. T3 –Data collection at Time 3, April 15–20, 2020. The above models were excellent fits for the data (e.g., CFI = 1.00) and were therefore retained.

**Reflection and Rumination Questionnaire RRQ; [33].** The 12-item scale measures individual differences in the use of repetitive, self-critical rumination. (e.g., "I spend a great deal of time thinking back over my embarrassing or disappointing moments"). Participants would rate their agreement with items on a scale of 1 (= strongly disagree) to 5 (= strongly agree). Subscale scores were calculated by summing together the items, with a possible range of 12–60, and higher scores indicating greater levels of rumination. The subscale showed excellent reliability at all time points (Cronbach's alphas = .96 at all time points).

**Distress Tolerance Scale [34].** The Distress Tolerance Scale utilizes 15 items to assess participants' preference for avoiding emotional distress (e.g., "I'll do anything to stop feeling distressed or upset"). Participants would rate their agreement with items on a scale of 1 (= strongly disagree) to 5 (= strongly agree). Subscale scores were calculated by summing together the items, with a possible range of 15–75, and higher scores indicating greater levels of distress intolerance. The subscale showed good reliability at all time points (Cronbach's alphas = .83–84).

**Coronavirus stressor items.** An ad-hoc questionnaire assessed participants' exposure to common stressors (e.g., "Job requires possible exposure to coronavirus"; See Table 1 for list of

**Table 1. Stressors related to COVID-19 and cross-sectional correlation with depression and anxiety.**

| Variable | Mean (SD) / Number in sample (Percent) | Depression T3 ($r$) | Anxiety T3 ($r$) |
|---|---|---|---|
| **Anxiety/stress as a result of COVID-19 pandemic** | 57.40 (28.00) | .38** | .32** |
| **Loneliness as a result of COVID-19 pandemic** | 42.33 (32.74) | .48** | .32** |
| **Became ill from possible exposure to COVID-19** | | | |
| Me | 7 (3.2%) | .16* | .15* |
| Close to me | 38 (17.4%) | .11 | .03 |
| n/a | 179 (82.1%) | -.10 | .03 |
| **Knows someone who died from COVID-19** | | | |
| Me | 9 (4.1%) | -.02 | .02 |
| Close to me | 18 (8.3%) | -.02 | -.00 |
| n/a | 194 (89.0%) | .02 | -.02 |
| **Job requires possible exposure to COVID-19** | | | |
| Me | 35 (16.1%) | .02 | .03 |
| Close to me | 58 (26.7%) | .02 | -.00 |
| n/a | 147 (67.4%) | -.06 | -.04 |
| **Lost job or reduced income due to COVID-19 pandemic** | | | |
| Me | 62 (28.2%) | .00 | -.02 |
| Close to me | 69 (31.7%) | .07 | -.01 |
| n/a | 121 (55.5%) | -.04 | -.02 |
| **Increased responsibilities at home due to COVID-19 pandemic** | | | |
| Me | 71 (32.3%) | .03 | .03 |
| Close to me | 41 (18.6%) | -.03 | .02 |
| n/a | 135 (61.8%) | -.02 | -.08 |
| **Self-isolating due to government regulation or recommendation** | | | |
| Me | 138 (63.3%) | .05 | .02 |
| Close to me | 95 (43.6%) | .11 | -.00 |
| n/a | 65 (29.4%) | -.12 | -.07 |
| **Currently living alone** | 45 (20.6%) | .07 | -.09 |

* indicates $p < .05$.

** indicates $p < .01$.

T3 –Data collection at Time 3, taking place from April 15-April 20, 2020.

items). Participants indicated whether such stressors have happened to them, to somebody close to them, or did not apply to either. Additionally, they used a slider to rate on a scale of 1 (= no stress/loneliness at all) to 100 (= a lot of anxiety/loneliness) the extent to which they experienced anxiety or stress due to the COVID-19 pandemic (i.e., COVID stress) and the extent to which they felt increased loneliness as a result of the pandemic (i.e., COVID loneliness).

## Procedure

Participants were recruited via the Prolific Academic Platform as part of an ongoing study on reinforcement sensitivity, emotion regulation, and affective psychopathology [35]. Five hundred and seventeen participants were initially recruited via the Prolific Academic platform on October 17, 2018 (i.e., T1). They completed a series of self-report questionnaires related to reinforcement sensitivity, emotion regulation and affective pathology followed by an unrelated behavioral task (e.g., for similar procedure, see [36]). Questionnaires only assessed recent levels of psychopathology. Histories of psychopathology or childhood risk factors (e.g., adverse childhood events) were not assessed. Participants included in the study had an approval rate of 95% or above following at least 50 completed tasks. Six months later, all participants who successfully completed the first study were invited to complete the same measures again, between the dates of April 15 and April 22, 2019 (i.e., T2). Three hundred and forty-eight participants (67.3% of T1) completed the study. This was generally consistent with attrition rates in other longitudinal Internet-based studies (e.g., 70% after three months [37]). Exactly one year later, participants who completed the T2 measures were contacted again, and 218 (62.6% of T2) completed the same study for a third time, between the dates of April 15, 2020 and April 20, 2020 (i.e., T3). Importantly, this assessment took place during a peak in COVID-19-related fatalities (Fig 2; [29]). Participants who returned for the final assessment showed small differences in symptom severity and moderate differences in age from those who did not (S1 File). Specifically, participants who returned were older than those who did not ($M = 41.90$, SD = 13.07 vs M = 34.07 SD = 10.28, $d = .65$, $p < .001$), less depressed ($M = 5.85$, SD = 5.64 vs $M = 7.76$, SD = 5.97, $d = .33$, $p = .004$), and less anxious ($M = 3,25$, SD = 3.91 vs $M = 4.33$, SD = 4.25, $d = .27$, $p = .019$). In order to minimize their effects on difference scores between timepoints, only participants who completed all three time-points were included in analyses.

## Analysis plan

Analytic strategies were selected based on the psychometric properties of the variables. Environmental stressors were measured with binary data and as such were summarized using frequency statistics (i.e., rate and percentage). The relationships between the environmental stressors and symptom severity were assessed using the recommended polychoric correlations [38]. Subjective measures of COVID-19 stress, rumination, distress intolerance, depression, and anxiety were measured using dimensional variables. As such, they were summarized using descriptive statistics and their interrelationships were calculated using Pearson's correlations. Polychoric correlations and Pearson's correlations were evaluated using the same significance cutoff of $p < .05$ and their effect sizes were considered comparable to each other.

Assessing the extent to which depression and anxiety changed or remained the same between time points was done in three steps. First, we performed conventional null hypothesis significance tests (NHST) to examine differences between time-points. Specifically, within-group *t* tests were used, with symptom and trait measures (e.g., DASS-Depression) entered as the repeated measures across consecutive time-points (i.e., T1-T2 or T2-T3). In this case, a significant finding would indicate that the repeated measures changed between time-points.

After observing the unexpectedly small effect sizes of change between time-points, we next estimated whether these effects were statistically equivalent to zero. This procedure is performed using a combination of the previous NHSTs and an additional two one-sided tests (TOST) procedure for equivalence testing [39]. In this procedure, an *a priori* smallest effect size of interest (SESOI) is calculated. Two one-sided *t*-tests then assess whether the full confidence interval of the change score estimate falls within the positive and negative SESOI (e.g., Cohen's $d$ = -.20 < X < .20). If so, the change score is judged as statistically equivalent to zero and scores are considered equivalent to each other. Finally, in the case of significant difference, we examined Cohen's $d$ of difference in order to assess the size of the difference between timepoints.

Finally, and most importantly, we compared two possible explanatory models for the relationships between COVID-19 stress, and loneliness and symptom severity. The models included binary data (i.e., environmental stressors) and were therefore calculated using polychoric correlations and a diagonal weighted least squares (DWLS) estimator [40]. Models were assessed using robust fit statistics using the recommended cutoffs [41, 42] of: non-significant chi-square test, CFI > .95, RMSEA < .06, SRMR < .08.

## Ethics

The study was conducted with the approval of the Ethics Committee of the Faculty of Social Sciences at the Hebrew University of Jerusalem (Approval #124120). All participants provided written consent prior to each wave of assessments.

## Results

Data and analysis syntax are available at https://osf.io/sjp4a/.

### Presence of stressors related to COVID-19

We first examined the pandemic's impact on our sample (see Table 1). Overall, we found that it introduced unique stressors to most participants' environments. Most participants (63.3%) were self-isolating as a result of government regulation. Similarly, 44.5% of the participants reported that either they or a person close to them has experienced a reduction of income as a result of the pandemic. Importantly, participants reported subjective feelings of distress as a result of the pandemic. They reported that, on average, it caused them moderate levels of loneliness ($M$ = 42.33, $SD$ = 32.74) and stress ($M$ = 57.40, $SD$ = 28.00; both on a 0–100 scale).

### Equivalence of measures across baselines

Next, we examined whether the measures of symptom severity and clinically relevant traits changed between the two baseline time-points (see Table 2; T1-T2), taken prior to the pandemic. We hypothesized that all measures would remain statistically equivalent on a group level. NHST *t*-tests revealed that depression, anxiety, and rumination were indeed non-different from each other (Cohen's $d$s = -.06 - .02, $p$s > .177). However, distress intolerance significantly decreased by a trivial degree, $t(217)$ = 1.98, $p$ = .049, $d$ = -.10 95% CI[-.19; .00], between T1 ($M$ = 42.94, SD = 10.26) and T2 ($M$ = 41.95, SD = 9.95).

In order to ascertain whether these small differences were statistically equal to zero, we performed a series of equivalence tests for each measure using the recommended TOST procedure [39]. Indeed, depression, anxiety, and rumination were all found to be significantly equivalent ($p$s ≤ .027). Thus, these three measures were judged to be equivalent and non-different across the two baseline timepoints. However, distress intolerance–which was found to

**Table 2. Equivalence testing of clinical measures and clinically relevant traits.**

| Comparison | | M (SD) | Cohen's d [90% CI] | NHST test for differences | TOST test for equivalence | Conclusion |
|---|---|---|---|---|---|---|
| **Depression** | | | | | | |
| | T1 | 5.94 (5.52) | -0.02 [-0.10; 0.06] | $t(217) = 0.4, p = .691$ | $t(217) = 2.89, p = .002$ | Equivalent and not different |
| | T2 | 5.85 (5.64) | | | | |
| | T2 | 5.85 (5.64) | 0.08 [-0.03; 0.18] | $t(217) = 1.45, p = .147$ | $t(217) = 4.74, p < .001$ | Equivalent and not different |
| | T3 | 6.28 (5.50) | | | | |
| **Anxiety** | | | | | | |
| | T1 | 3.17 (3.92) | 0.02 [-0.07; 0.12] | $t(217) = 0.43, p = .665$ | $t(217) = 3.72, p < .001$ | Equivalent and not different |
| | T2 | 3.25 (3.91) | | | | |
| | T2 | 3.25 (3.91) | -0.11 [-0.22; 0.00] | $t(217) = 1.98, p = .048$ | $t(217) = 1.30, p = .097$ | Not equivalent and different |
| | T3 | 2.83 (3.61) | | | | |
| **Rumination** | | | | | | |
| | T1 | 43.11 (11.02) | -0.06 [-0.14; 0.03] | $t(217) = 1.35, p = .177$ | $t(217) = 1.94, p = .027$ | Equivalent and not different |
| | T2 | 42.46 (11.74) | | | | |
| | T2 | 42.46 (11.74) | -0.07 [-0.15; 0.02] | $t(217) = 1.59, p = .112$ | $t(217) = 1.69, p = .046$ | Equivalent and not different |
| | T3 | 41.66 (12.15) | | | | |
| **Distress Tolerance** | | | | | | |
| | T1 | 42.94 (10.26) | -0.10 [-0.19; 0.00] | $t(217) = 1.98, p = .049$ | $t(217) = 1.31, p = .096$ | Not equivalent and different |
| | T2 | 41.95 (9.95) | | | | |
| | T2 | 41.95 (9.95) | -0.09 [-0.19; 0.01] | $t(217) = 1.78, p = .076$ | $t(217) = 2.86, p = .002$ | Equivalent and not different |
| | T3 | 41.03 (10.08) | | | | |

Note. T1 –Data collection at Time 1, October 17, 2018. T2 –Data collection at Time 2, April 15–22, 2019. T3 –Data collection at Time 3, April 15–20, 2020.

be significantly different from T1 to T2 using NHST ($d = .10$, $p = .049$)–was not found to be significantly equivalent using TOST ($p = .096$). As such, distress intolerance was judged to be different between T1 and T2, to a significant albeit trivial degree.

## Equivalence between baseline and COVID-19 pandemic

Next, to assess the validity of a general-stressor model, we examined whether any group-level change occurred between the second baseline assessment and the assessment that occurred during the height of the COVID-19 pandemic one year later (see Table 2; T2-T3). We expected to find increases in all these distress-related measures and tested this hypothesis with a series of within-group $t$-tests to detect differences between timepoints. Contrary to our hypothesis, no change was detected for depression, rumination or distress tolerance ($d$s = -.09 - .08, $p$s > .076). While significant change did occur for anxiety, the effect was trivially small, and similar to the other measures that did not show significant change, $t(217) = 1.98$, $p = .048$, $d = -.11$, 95% CI [-.22; .00]. More importantly, this trivial change occurred in the direction opposite from what was hypothesized with anxiety during the pandemic (T3 $M = 2.83$, SD = 3.61) being slightly lower than at baseline (T2 $M = 3.25$, SD = 3.91).

Due to the non-significant findings for depression, rumination, and distress intolerance, and the small effect size found for anxiety, we performed a series of post-hoc equivalence tests to assess whether these effect sizes were statistically equivalent to zero. Indeed, depression, rumination and distress intolerance were equivalent between baseline and during the COVID-19 pandemic ($d$s = -.09 - .08, $p$s < .046). Anxiety, on the other hand, was found to be non-equivalent between baseline and during the pandemic, $t(217) = 1.3$, $p = .097$, $d = -.11$, 95% CI

[-.22; .00]. Thus, depression, rumination, and distress intolerance were equivalent to baseline, while a trivial albeit significant decrease in anxiety was observed as well.

## Relationship between COVID-19 stress and symptom severity

After we did not find a group-level increase in depression or anxiety, we next examined the relationship between pandemic-related distress and psychopathology at the onset of the pandemic. First, we examined the bivariate correlations between stressors related to COVID-19, depression, and anxiety (see Table 1). Previous illness from possible exposure to COVID-19 was the only environmental stressor that was significantly associated with depression ($r = .16$) and anxiety ($r = .15$). No other environmental stressor significantly correlated with depression or anxiety ($ps = .094 - .986$). On the other hand, pandemic-related distress (i.e., loneliness and stress as a result of COVID-19) was associated with both depression ($r = .42$ and $.38$, $p < .001$, respectively) and anxiety ($rs = .32$, $p < .001$).

We then performed our primary analysis, comparing two explanatory models for how pandemic-related distress related to clinical symptoms. In the general stressor model (Fig 2a and 2b), pandemic-related environmental stressors predicted change in psychopathology. This model included the hypotheses that (a) environmental stressors should predict emotional distress, and (b) that emotional distress should predict symptom severity beyond baseline. Due to the relationship between social disconnect and depression [23], the first model specified self-isolation to predict COVID-related loneliness which in turn predicted depression after controlling for baseline (Fig 3a). Due to the relationship between economic uncertainty and anxiety [8], the second model specified negative effects on participants' incomes to predict COVID-related anxiety/stress, which in turn predicted anxiety after controlling for baseline (Fig 3b). These models fit the data very poorly (see Table 3).

In the diathesis-stress model (Fig 4a and 4b), both psychopathology and environmental stressors independently predicted current levels of COVID-related distress. In these models, (a) only baseline measures predicted symptom severity at T3 and emotional distress was independently predicted by (b) symptom severity at T3 (c) as well as environmental stressors. Thus, for depression (Fig 4a), we specified depression at T2 to predict depression at T3. COVID-related feelings of loneliness were predicted by depression at T3 and whether the participant was self-isolating. For anxiety (Fig 4b), anxiety at T2 predicted anxiety at T3, which then predicted subjective COVID-related stress along with loss of income. These models were excellent fits for the data (see Table 3).

Taken together, we found that the COVID-19 pandemic did not lead to a general increase in depression or anxiety. Furthermore, distress caused by environmental stressors did not lead to a rise in symptom severity. Rather, participants' ratings of COVID-19-related distress were independently predicted by symptom severity and environmental stressors. Thus, instead of a

**Table 3. Summary of model fit statistics for the alternative path models.**

| Model | $\chi^2$ (*df*) | *p* value | Robust CFI | Robust RMSEA | RMSEA 90% CI | Robust SRMR |
|---|---|---|---|---|---|---|
| **Depression models** | | | | | | |
| **Loneliness -> depression model** (Fig 3a) | 28.60 (2) | < .001 | .000 | .248 | .172-.332 | .002 |
| **Depression -> loneliness model** (Fig 4a) | .177 (2) | .92 | 1.00 | 0.00 | 0.00–0.50 | .010 |
| **Anxiety models** | | | | | | |
| **Stress -> anxiety model** (Fig 3b) | 16.25 (2) | < .001 | 0.00 | .181 | .107-.267 | .002 |
| **Anxiety -> stress model** (Fig 4b) | 1.08 (2) | .583 | 1.00 | .000 | .000-.011 | .034 |

CFI = comparative fit index; RMSEA = root-mean-square error of approximation; CI = confidence interval; SRMR = Standardized root mean square residual.

general-stressor model wherein the pandemic caused a group-level rise in symptom severity, a diathesis-stress model was supported, wherein prior symptom severity predicted pandemic-related distress.

## Discussion

The current study models the relationship between the COVID-19 pandemic, distress, depression, and anxiety during the pandemic's early stages. It did so by longitudinally assessing participants' levels of depression, anxiety, rumination and distress intolerance from two baseline assessments 12 and 18 months prior to the pandemic's high points in Spring 2020. Prior to analysis, we expected to observe a rise in depression and anxiety, resulting from distress associated with environmental, pandemic-related stressors [10, 12]. However, no rise occurred. Instead, symptoms and clinically relevant measures were either equivalent or reduced by a trivial degree in the case of anxiety. The only non-trivial predictor of symptom severity during the pandemic was symptom severity at baseline.

While unexpected, these findings were consistent with patterns observed following other disasters. Populations appear to be resilient to large-scale environmental stressors [20]. As such, a broad, simplistic general-stressor model of the COVID-19 pandemic has not been supported in the current study or elsewhere [18]. Rather, the current study argues that diathesis-stress models [22, 43] are necessary to identify how and for whom pandemic-related stressors may lead to psychopathology.

The current study also identifies individuals who may be more at risk for distress during the COVID-19 pandemic. After comparing alternative explanatory models for the relationship between environmental stressors, subjective distress, and symptom severity, we found that the model with an excellent fit for the data specified distress to be independently predicted by environmental stressors and psychopathology. The alternate model that specified psychopathology to be predicted by distress [10] fit the data very poorly. Indeed, the only environmental stressor to predict symptom severity was possible previous infection with COVID-19 [14]. Thus, those most at risk for distress during the pandemic were those experiencing both environmental stressors relevant to the distress (e.g., economic stress) as well as a history of mental illness (e.g., anxiety). These findings may complement others that use demographic correlates to identify those at higher risk for elevated mental illness during the pandemic. Factors such as age, gender, race, ethnicity, location, and education are associated with internalizing pathology at the onset of the pandemic, alongside pandemic-related stressors [44]. These different risk factors interact syndemically, wherein pandemic-related stressors have greater impacts upon mental health for those with minoritized racial identities [45] or live in lower-income communities [46]. While the current study's sample size is too small to undertake such interaction analyses, other large-scale studies will be better equipped to do so.

The current study highlights the importance of including baseline measures when studying the effects of the COVID-19 pandemic [13, 27]. One of its key strengths is the inclusion of two assessments, occurring exactly 12 months and 18 months prior. This context highlights the lack of meaningful difference in the sample's clinical measures, even when the pandemic was underway. Future research on the COVID-19 pandemic's impact will ideally include baseline assessments of clinical measures from prior to the pandemic's stressors were introduced [11]. These baseline measures are often missing from disaster research [20] and likewise should be emphasized as a key limitation to a cross-sectional study's ability to directly assess the role of COVID-19 as a stressor. Indeed, it also more generally highlights the importance of longitudinal data in identifying those most at risk for elevated levels of mental illness during periods of great distress. For example, individuals with adverse childhood experiences may also be at risk

for increased levels of mental illness during times of pandemic-related stress [47]. In this case, ongoing, lifetime longitudinal studies are specially equipped to assess the relationship between diatheses related to earlier experiences and later distress [16].

Future research may limit or replicate the current study's findings by comparing varied methods of sampling and assessment when measuring symptom severity. For example, the current study utilized an Internet-based sample. While such samples show similar rates of depression and anxiety as the general population [48], they may also differ from in terms of traits relevant to self-isolation such as lower extraversion and higher Internet use [49]. It is also possible that the assessment that took place during the COVID-19 pandemic (i.e., T3) did not occur at the most distressing time for participants. Although it occurred during peak rates of COVID-19 fatality (see Fig 2), other events during the pandemic may have led to other distress peaks as well, such as when participants began to self-isolate [50].

Additionally, the differences between those who returned for the final assessment and those who did not is worthy of acknowledgement. Although the differences in symptom severity were small and of low clinical significance before the pandemic, it is possible that the participants who did not return for the final assessment were more at risk for symptom increase than those who did. In such a case, it is possible that had they been included, equivalence across timepoints may not have occurred. This hypothetical circumstance, however, is also inconsistent with the general stressor model that predicts symptom increase across the population. Furthermore, the current study is consistent with other large-scale studies that have also not found that symptom severity increased precipitously during the pandemic [6, 18, 19]. Further research may benefit from focusing on how the pandemic impacted well-being among participants at different levels of symptom severity.

Finally, the current study approaches risk factors for mental illness from the perspectives of contemporary intrapersonal processing, acute stress, and internalizing psychopathology. However, many risk factors for elevated mental illness occurred prior to the original baseline measures included in the study. Elevated levels of psychopathology prior to assessment [16] or adverse childhood events [47] are both useful for identifying those at risk for greater levels of distress while facing pandemic-related stressors. Future studies using similar designs may use retrospective assessments to capture some of these risk factors as well. Similarly, many risk factors both preceding and during the COVID-19 pandemic are interpersonal in nature. Social stressors both from the participant's past (e.g., and during their experiences in quarantine (e.g., domestic violence) [51] are themselves likely risk factors for loneliness, and mood disorder [52]. Future studies may further examine the role of interpersonal factors in well-being at the onset of the pandemic [47]. Furthermore, the current research design focuses on the acute periods of stress that accompanied the beginning of the pandemic. However, as the pandemic continued, stress has accumulated. Future studies may assess the association between cumulative stress and mental illness [47]. Additionally, the current study operationalized mental health in terms of metrics of distress, loneliness, depression, and anxiety. However, other metrics of behavioral health, such as substance use and behavioral addictions, have also been associated with social isolation during the pandemic's onset [9]. The current study may be considered against others related to externalizing disorders in order to assess trends across different types of psychopathology.

Collaborative work between multiple laboratories will also offer opportunities to compare findings among diverse populations, assessment methods, and clinical histories in order to more efficiently identify at-risk populations [53]. Identifying which participants are most vulnerable to mental illness in the face of pandemic-related stress is the first step in developing targeted interventions [54]. In doing so, the clinical science community may rise to the formidable challenge that COVID-19 has set before it.

## Supporting information

**S1 Table. Additional demographic information of sample.**
(DOCX)

**S2 Table. Comparison of participants that did and did not return for later assessments.** T1 returners are those who returned for T2. T1 dropouts are those who did not return for T2. T2 returners are those who completed T1, T2, and T3. T2 returners are those who completed T1 and T2 but not T3.
(DOCX)

**S1 File. Main analyses including only participants from the United Kingdom and United States.**
(DOCX)

## Author Contributions

**Conceptualization:** Benjamin A. Katz, Iftah Yovel.

**Data curation:** Benjamin A. Katz.

**Formal analysis:** Benjamin A. Katz.

**Funding acquisition:** Iftah Yovel.

**Investigation:** Benjamin A. Katz.

**Methodology:** Benjamin A. Katz, Iftah Yovel.

**Project administration:** Benjamin A. Katz.

**Resources:** Benjamin A. Katz.

**Software:** Benjamin A. Katz.

**Supervision:** Iftah Yovel.

**Visualization:** Benjamin A. Katz.

**Writing – original draft:** Benjamin A. Katz.

**Writing – review & editing:** Benjamin A. Katz, Iftah Yovel.

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
