## [Editor Report · Decision Letter 0]

17 Sep 2021

PONE-D-21-18866Mood Symptoms Predict COVID-19 Pandemic Distress but not Vice Versa:An 18-Month Longitudinal StudyPLOS ONE

Dear Dr. Katz,

Thank you for submitting your manuscript to PLOS ONE. After careful consideration, we feel that it has merit but does not fully meet PLOS ONE’s publication criteria as it currently stands. Therefore, we invite you to submit a revised version of the manuscript that addresses the points raised during the review process.Please submit your revised manuscript by Nov 01 2021 11:59PM. If you will need more time than this to complete your revisions, please reply to this message or contact the journal office at plosone@plos.org. Please include the following items when submitting your revised manuscript:A rebuttal letter that responds to each point raised by the academic editor and reviewer(s). You should upload this letter as a separate file labeled 'Response to Reviewers'.A marked-up copy of your manuscript that highlights changes made to the original version. You should upload this as a separate file labeled 'Revised Manuscript with Track Changes'.An unmarked version of your revised paper without tracked changes. You should upload this as a separate file labeled 'Manuscript'.If applicable, we recommend that you deposit your laboratory protocols in protocols.io to enhance the reproducibility of your results. Protocols.io assigns your protocol its own identifier (DOI) so that it can be cited independently in the future. For instructions see: https://journals.plos.org/plosone/s/submission-guidelines#loc-laboratory-protocols. Additionally, PLOS ONE offers an option for publishing peer-reviewed Lab Protocol articles, which describe protocols hosted on protocols.io. Read more information on sharing protocols at https://plos.org/protocols?utm_medium=editorial-email&utm_source=authorletters&utm_campaign=protocols.

We look forward to receiving your revised manuscript.

Kind regards,

Vedat Sar, M.D.

Academic Editor

PLOS ONE
---

## [Author Response · Author response to Decision Letter 0]

29 Sep 2021

Editor’s comments

1. Please ensure that your manuscript meets PLOS ONE's style requirements, including those for file naming

We have implemented the following changes consistent with PLOS ONE’s style book:

• Headings, table titles and figure titles were adjusted in terms of location and font size.

• Citations were changed to PLOS ONE style instead of APA style.

• Tables and figure titles were integrated into the text of the manuscript and Supplementary Table 1 was noted at the end.

2. Please update your submission to use the PLOS LaTeX template

In keeping with PLOS ONE policy, we uploaded a .docx version of the manuscript with and without changes tracked.

The following ethics statement is now included at the end of the Method section:

The study was conducted with the approval of the Ethics Committee of the Faculty of Social Sciences at the Hebrew University of Jerusalem under the study, "A path model for the connection between reinforcement sensitivity theory, emotion regulation, and psychopathology". All participants provided written consent prior to each wave of assessments.

Additionally, all figures were converted to .tif format and passed through the Preflight Analysis and Conversion Engine (PACE) digital diagnostic tool prior to upload. We have also carefully followed all file naming conventions as requested.

---

## [Decision Letter · Decision Letter 1]

18 May 2022

PONE-D-21-18866R1Mood Symptoms Predict COVID-19 Pandemic Distress but not Vice Versa: An 18-Month Longitudinal StudyPLOS ONE

Dear Dr. Katz,

Thank you for submitting your manuscript to PLOS ONE. After careful consideration, we feel that it has merit but does not fully meet PLOS ONE’s publication criteria as it currently stands. Therefore, we invite you to submit a revised version of the manuscript that addresses the points raised during the review process.

We look forward to receiving your revised manuscript.

Kind regards,

Mohammad Farris Iman Leong Bin Abdullah, Dr Psych

Academic Editor

PLOS ONE

Journal Requirements:

Additional Editor Comments (if provided):

Reviewers' comments:

Reviewer's Responses to Questions

**Comments to the Author**

1. If the authors have adequately addressed your comments raised in a previous round of review and you feel that this manuscript is now acceptable for publication, you may indicate that here to bypass the “Comments to the Author” section, enter your conflict of interest statement in the “Confidential to Editor” section, and submit your "Accept" recommendation.

Reviewer #1: (No Response)

Reviewer #2: (No Response)

2. Is the manuscript technically sound, and do the data support the conclusions?

Reviewer #1: Yes

Reviewer #2: Partly

3. Has the statistical analysis been performed appropriately and rigorously? 

Reviewer #1: I Don't Know

Reviewer #2: Yes

4. Have the authors made all data underlying the findings in their manuscript fully available?

Reviewer #1: No

Reviewer #2: Yes

5. Is the manuscript presented in an intelligible fashion and written in standard English?

Reviewer #1: Yes

Reviewer #2: Yes

6. Review Comments to the Author

Reviewer #1: p. 3

“The high levels of health and financial uncertainly salient to the pandemic are strongly linked to internalizing pathology [3].”

What about “externalizing pathology”. Seems important to reference as well.

p. 4

“Thus, in order to assess the COVID-19 pandemic’s negative impact on mental health, it is necessary to identify participants with pre-pandemic baseline data available, and to separately assess clinical symptoms, environmental stressors, and subjective distress.”

Thoughtful, complex design.

…

“The pandemic assessment period occurred between April 15, 2020 and April 20, 2020,..”

This research is limited by the lack of measurements 6 or 12 or 18 months into the pandemic, that reflects the effects of long-term, cumulative stress.

p. 5-6

Where are the demographic data. It appears pre-disaster trauma history (e.g. ACES study) was not included. Given that interpersonal trauma could be a predictor of post-disaster/pandemic reactivity, this should at minimum be mentioned as a study limitation.

p. 6

“They represented a wide range of ages (M = 42.87, SD = 13.09, range = 19 – 75)”

It would be helpful to see the data on how many people in each age grouping, example 19-29, 30-39, etc. to see if age cohorts mattered or not.

p. 9-10

“The relationships between the environmental stressors and symptom severity were assessed using polychoric correlations. All other relationships (e.g., between subjective COVID-19 stress and symptom severity) were calculated using Pearson’s correlations.”

I am not well-versed in statistics, which perhaps is why it would be helpful for the reader to understand why two different measurements of correlation were chosen, and what they each differentially show and don’t show.

…

“After observing the unexpectedly small effect sizes of change between time-points, we next estimated whether these effects were statistically equivalent to zero. This was done using the two one-sided tests (TOST) procedure for equivalence testing [26].”

It is my understanding the more tests you run, the more you’re likely to “find something” of statistical significance. Is this taken into account? If so, specify how.

p. 13

“Distress intolerance, on the other hand, was not found to be significantly equivalent (p = .096) across timepoints. As such, distress intolerance was judged to be different between T1 and T2, to a significant albeit trivial degree.”

Is p= .096 considered significant?? Shouldn’t it be lower than .05? Perhaps this should be explained as to why this is statistically significant, however “trivial”.

p. 17

“While unexpected, these findings were consistent with patterns observed following other disasters. Populations appear to be resilient to large-scale environmental stressors [9].”

How much is this a function of this being very early in the pandemic? Would it change after 1 year? After getting the vaccine, and then new variants causing a resurgence?

Also, the subject demographics does not assessment of interpersonal trauma history, e.g, along the lines of the ACES studies. In my clinical experience, the trauma history pre-“disaster” has a significant effect on post-disaster coping. At the minimum, the fact that interpersonal trauma history was NOT assessed should be mentioned.

…

“…those experiencing both environmental stressors relevant to the distress (e.g., economic stress) as well as a history of mental illness (e.g., anxiety).”

Again, trauma history is not included. Some of the anxiety, depression, isolation, etc. could be a consequence of post-trauma sequelae. Again, this limitation of the study needs mentioning.

p. 18

“Although it occurred during peak rates of COVID-19 fatality (see Figure 1), other events during the pandemic may have led to other distress peaks as well, such as when participants began to self-isolate [33].”

This is a good point.

..

“Identifying which participants are most vulnerable to mental illness in the face of pandemic-related stress is the first step in developing targeted interventions [35]. In doing so, the clinical science community may rise to the formidable challenge that COVID-19 has set before it.”

Agreed, but I don’t think the researchers take into account what I believe (and I assume research has shown) the importance of interpersonal, particularly severe childhood trauma history (a la the ACES research) and neglect. If you don’t take into account childhood trauma and neglect, your interventions will not be as patient-specific and effective as desired.

Reviewer #2: I appreciate the authors' hard efforts in this investigation. The submitted work has the following strengths: (i) a longitudinal design that can provide scientific evidence in temporal association; (ii) robust statistical analyses that can examine the theories proposed in the present study; (iii) the use of theoretical model for investigation. However, there are some concerns in the present work and the authors are encouraged to revise their work according to the following comments.

1. The Introduction should include some systematic reviews reporting the evidence of mental health issues during COVID-19 pandemic to emphasize the importance to investigate mental health during COVID-19 pandemic. Please see the following suggestions.

Rajabimajd, N., Alimoradi, Z., & Griffiths, M. D. (2021). Impact of COVID-19-related fear and anxiety on job attributes: A systematic review. Asian Journal of Social Health and Behavior, 4, 51-55

Olashore, A. A., Akanni, O. O., Fela-Thomas, A. L., & Khutsafalo, K. (2021). The psychological impact of COVID-19 on health-care workers in African Countries: A systematic review. Asian Journal of Social Health and Behavior, 4, 85-97

Alimoradi, Z., Ohayon, M. M., Griffiths, M. D., Lin, C.-Y., & Pakpour, A. H. (2022). Fear of COVID-19 and its association with mental health related factors: A systematic review and meta-analysis. BJPsych Open, 8, e73.

Alimoradi, Z., Lin, C.-Y., Ullah, I., Griffiths, M. D., & Pakpour, A. H. (2022). Item response theory analysis of Fear of COVID-19 Scale (FCV-19S): A systematic review. Psychology Research and Behavior Management, 15, 581-596.

Alimoradi, Z., Gozal, D., Tsang, H. W. H., Lin, C.-Y., Broström, A., Ohayon, M. M., & Pakpour, A. H. (2022). Gender-specific estimates of sleep problems during the COVID-19 pandemic: Systematic review and meta-analysis. Journal of Sleep Research, 31(1), e13432.

Alimoradi, Z., Broström, A., Tsang, H. W. H., Griffiths, M. D., Haghayegh, S., Ohayon, M. M., Lin, C.-Y., Pakpour, A. H. (2021). Sleep problems during COVID-19 pandemic and its’ association to psychological distress: A systematic review and meta-analysis. EClinicalMedicine, 36, 100916.

2. The authors tested two theoretical modes (i.e., general-stressor model and diathesis-stressor model). However, they did not introduce the two models in the Introduction. They only briefly mention the general concepts of the two models. However, given that the study’s main focus is to examine and compare the two models, the authors should elaborate the information and descriptions of the two models. It would be much better if the authors also use the figures to explain the two models in the Introduction.

3. The majority of the participants were recruited from the US and the UK. Also, the authors described more COVID-19 information for the two countries. Therefore, I think that it is necessarily to do a sensitivity analysis on the US and UK samples (i.e., removing the Canada and Ireland participants to examined the tested models).

4. The description of DASS is unclear. From the description of “seven-item subscales”, I know that the authors used DASS-21 instead of DASS-42. However, the authors did not make it clear that they have used DASS-21. Moreover, the citation credit should give to Lovibond and Lovibond (1995), as they are the original developers. It is fine to cite Henry and Crawford. However, Lovibond and Lovibond cannot be excluded in the citations.

5. The authors should provide scoring information for all the instruments. Also, the meaning of the directions in each instrument should be provided. Without such information, one cannot interpret the scores for the instruments.

6. From the Procedure section, one can understand that the attrition rate of the longitudinal study was about 40%. This is fine. However, the authors should provide information regarding whether the retained participants and the lost-to-follow-up participants share similar demographics. This can check if the missing is at random.

7. From the statement, “Participants were recruited via the Prolific Academic Platform as part of an ongoing study on reinforcement sensitivity, emotion regulation, and affective psychopathology”, I wonder whether the authors have some publications already published to make a citation here.

8. Tables 2 and 3 are out of the size and cannot be read.

9. For all the tables, the authors should use footnotes to provide definition of T1, T2, and T3.

7. PLOS authors have the option to publish the peer review history of their article (what does this mean?). If published, this will include your full peer review and any attached files.

Reviewer #1: No

Reviewer #2: No

---

## [Author Response · Author response to Decision Letter 1]

20 Jun 2022

Reviewer #1: 

p. 3

“The high levels of health and financial uncertainly salient to the pandemic are strongly linked to internalizing pathology [3].”

What about “externalizing pathology”. Seems important to reference as well.

We have now included a citation for externalizing pathology in the sentence (p. 3):

The high levels of health and financial uncertainly salient to the pandemic are strongly linked to stress, [7] internalizing pathology [8] and externalizing pathology [9].

p. 4

“The pandemic assessment period occurred between April 15, 2020 and April 20, 2020,..”

This research is limited by the lack of measurements 6 or 12 or 18 months into the pandemic, that reflects the effects of long-term, cumulative stress.

We have now included this point in the limitation section (pp. 23):

Furthermore, the current research design focuses on the acute periods of stress that accompanied the beginning of the pandemic. However, as the pandemic continued, stress has accumulated. Future studies may assess the association between cumulative stress and mental illness [47]. 

p. 5-6

Where are the demographic data. It appears pre-disaster trauma history (e.g. ACES study) was not included. Given that interpersonal trauma could be a predictor of post-disaster/pandemic reactivity, this should at minimum be mentioned as a study limitation.

Demographic data, including gender, location, race/ethnicity, age, education, and employment status, are available in S1 Table (p. 29). We have checked prior to uploading that this table is available for review.

Additionally, we now include the lack of assessment of adverse childhood experiences and other interpersonal factors as a limitation on p. 23:

Finally, the current study approaches risk factors for mental illness from the perspectives of intrapersonal processing and of acute stress. However, many risk factors both preceding and during the COVID-19 pandemic are interpersonal in nature. Social stressors both from the participant’s past (e.g., adverse childhood events) [47] and during their experiences in quarantine (e.g., domestic violence) [48] are themselves likely risk factors for loneliness, and mood disorder [49]. Future studies may further examine the role of interpersonal factors in well-being at the onset of the pandemic [47].

p. 6

“They represented a wide range of ages (M = 42.87, SD = 13.09, range = 19 – 75)”

It would be helpful to see the data on how many people in each age grouping, example 19-29, 30-39, etc. to see if age cohorts mattered or not.

This breakdown of ages is now included in the demographics table, S1 Table.

p. 9-10

“The relationships between the environmental stressors and symptom severity were assessed using polychoric correlations. All other relationships (e.g., between subjective COVID-19 stress and symptom severity) were calculated using Pearson’s correlations.”

I am not well-versed in statistics, which perhaps is why it would be helpful for the reader to understand why two different measurements of correlation were chosen, and what they each differentially show and don’t show.

Response:

To clarify this issue, we have now rewritten the beginning of the analysis plan to explain the use of polychoric vs Pearson correlations as well as how to interpret them (p. 11):

Analytic strategies were selected based on the psychometric properties of the variables. Environmental stressors were measured with binary data and as such were summarized using frequency statistics (i.e., rate and percentage). The relationships between the environmental stressors and symptom severity were assessed using the recommended polychoric correlations [38]. Subjective measures of COVID-19 stress, rumination, distress intolerance, depression, and anxiety were measured using dimensional variables. As such, they were summarized using descriptive statistics and their interrelationships were calculated using Pearson’s correlations. Polychoric correlations and Pearson’s correlations were evaluated using the same significance cutoff of p < .05 and their effect sizes were considered comparable to each other..

…

“After observing the unexpectedly small effect sizes of change between time-points, we next estimated whether these effects were statistically equivalent to zero. This was done using the two one-sided tests (TOST) procedure for equivalence testing [26].”

It is my understanding the more tests you run, the more you’re likely to “find something” of statistical significance. Is this taken into account? If so, specify how.

Response:

We certainly agree with the reviewer’s concern regarding degrees of freedom that researchers take for themselves. In keeping with this concern, we follow the Open Science movement approach that recommends transparent reporting of which analyses were chosen a priori and which were secondary (e.g., Benning et al., 2019). In this case, we found the data to be more appropriate for a TOST procedure, which incorporates both null hypotheses significance testing (NHST) and TOST. Importantly, we did not perform other tests on these data germane to the current study, and have now included a citation when describing the study that may refer interested readers to the other tests that have been performed on this dataset in a separate study (p. 10):

Participants were recruited via the Prolific Academic Platform as part of an ongoing study on reinforcement sensitivity, emotion regulation, and affective psychopathology [35] 

Source: Benning, S. D., Bachrach, R. L., Smith, E. A., Freeman, A. J., & Wright, A. G. (2019). The registration continuum in clinical science: A guide toward transparent practices. Journal of Abnormal Psychology, 128(6), 528.

p. 13

“Distress intolerance, on the other hand, was not found to be significantly equivalent (p = .096) across timepoints. As such, distress intolerance was judged to be different between T1 and T2, to a significant albeit trivial degree.”

Is p= .096 considered significant?? Shouldn’t it be lower than .05? Perhaps this should be explained as to why this is statistically significant, however “trivial”.

Response:

All differences in time-points were subjected to two rounds of testing. First, we used null-hypothesis significance testing (NHST) to evaluate difference at the two time points. Then, we used the two one-sided tests (TOST) approach to examine equivalence. A significant NHST indicates difference, while a significant TOST indicates equivalence. In this case, NHST of distress tolerance from T1 to T2 previously found significant difference (d = -.10, p = .049). Then, at the point in analysis the reviewer references, we observed a failed TOST (p = .096), indicating non-equivalence. We thus concluded that the values were different, based on NHST, and non-equivalent, based on TOST.

To clarify this point, we have now updated the text of that section of analysis to clarify this difference (p. 13):

However, distress intolerance – which was found to be significantly different from T1 to T2 using NHST (d = .10, p = .049) – was not found to be significantly equivalent using TOST (p = .096).

p. 17

“While unexpected, these findings were consistent with patterns observed following other disasters. Populations appear to be resilient to large-scale environmental stressors [9].”

How much is this a function of this being very early in the pandemic? Would it change after 1 year? After getting the vaccine, and then new variants causing a resurgence?

Also, the subject demographics does not assessment of interpersonal trauma history, e.g, along the lines of the ACES studies. In my clinical experience, the trauma history pre-“disaster” has a significant effect on post-disaster coping. At the minimum, the fact that interpersonal trauma history was NOT assessed should be mentioned.

…

“…those experiencing both environmental stressors relevant to the distress (e.g., economic stress) as well as a history of mental illness (e.g., anxiety).”

Again, trauma history is not included. Some of the anxiety, depression, isolation, etc. could be a consequence of post-trauma sequelae. Again, this limitation of the study needs mentioning.

Response:

We now include these as limitations to the study (p. 23):

Finally, the current study approaches risk factors for mental illness from the perspectives of intrapersonal processing and of acute stress. However, many risk factors both preceding and during the COVID-19 pandemic are interpersonal in nature. Social stressors both from the participant’s past (e.g., adverse childhood events) [47] and during their experiences in quarantine (e.g., domestic violence) [48] are themselves likely risk factors for loneliness, and mood disorder [49]. Future studies may further examine the role of interpersonal factors in well-being at the onset of the pandemic [47]. Furthermore, the current research design focuses on the acute periods of stress that accompanied the beginning of the pandemic. However, as the pandemic continued, stress has accumulated. Future studies may assess the association between cumulative stress and mental illness [47].

p. 18

“Identifying which participants are most vulnerable to mental illness in the face of pandemic-related stress is the first step in developing targeted interventions [35]. In doing so, the clinical science community may rise to the formidable challenge that COVID-19 has set before it.”

Agreed, but I don’t think the researchers take into account what I believe (and I assume research has shown) the importance of interpersonal, particularly severe childhood trauma history (a la the ACES research) and neglect. If you don’t take into account childhood trauma and neglect, your interventions will not be as patient-specific and effective as desired.

We now mention adverse childhood events and other interpersonal risk factors in the new limitation paragraph (see above response). 

Reviewer #2

1. The Introduction should include some systematic reviews reporting the evidence of mental health issues during COVID-19 pandemic to emphasize the importance to investigate mental health during COVID-19 pandemic. Please see the following suggestions.

Rajabimajd, N., Alimoradi, Z., & Griffiths, M. D. (2021). Impact of COVID-19-related fear and anxiety on job attributes: A systematic review. Asian Journal of Social Health and Behavior, 4, 51-55

Olashore, A. A., Akanni, O. O., Fela-Thomas, A. L., & Khutsafalo, K. (2021). The psychological impact of COVID-19 on health-care workers in African Countries: A systematic review. Asian Journal of Social Health and Behavior, 4, 85-97

Alimoradi, Z., Ohayon, M. M., Griffiths, M. D., Lin, C.-Y., & Pakpour, A. H. (2022). Fear of COVID-19 and its association with mental health related factors: A systematic review and meta-analysis. BJPsych Open, 8, e73.

Alimoradi, Z., Lin, C.-Y., Ullah, I., Griffiths, M. D., & Pakpour, A. H. (2022). Item response theory analysis of Fear of COVID-19 Scale (FCV-19S): A systematic review. Psychology Research and Behavior Management, 15, 581-596.

Alimoradi, Z., Gozal, D., Tsang, H. W. H., Lin, C.-Y., Broström, A., Ohayon, M. M., & Pakpour, A. H. (2022). Gender-specific estimates of sleep problems during the COVID-19 pandemic: Systematic review and meta-analysis. Journal of Sleep Research, 31(1), e13432.

Alimoradi, Z., Broström, A., Tsang, H. W. H., Griffiths, M. D., Haghayegh, S., Ohayon, M. M., Lin, C.-Y., Pakpour, A. H. (2021). Sleep problems during COVID-19 pandemic and its’ association to psychological distress: A systematic review and meta-analysis. EClinicalMedicine, 36, 100916.

We have now included updated citations from this list and others, including: 

Alimoradi Z, Gozal D, Tsang HWH, Lin C, Broström A, Ohayon MM, et al. Gender‐specific estimates of sleep problems during the COVID‐19 pandemic: Systematic review and meta‐analysis. Journal of Sleep Research. 2022;31. doi:10.1111/jsr.13432

Alimoradi Z, Ohayon MM, Griffiths MD, Lin C-Y, Pakpour AH. Fear of COVID-19 and its association with mental health-related factors: systematic review and meta-analysis. BJPsych Open. 2022;8: e73. doi:10.1192/bjo.2022.26

Avena NM, Simkus J, Lewandowski A, Gold MS, Potenza MN. Substance use disorders and behavioral addictions during the COVID-19 pandemic and COVID-19-related restrictions. Frontiers in Psychiatry. 2021;12. doi:10.3389/fpsyt.2021.653674

Berman NC, Fang A, Hoeppner SS, Reese H, Siev J, Timpano KR, et al. COVID-19 and obsessive-compulsive symptoms in a large multi-site college sample. Journal of Obsessive-Compulsive and Related Disorders. 2022; 100727. doi:10.1016/j.jocrd.2022.100727

Bourmistrova NW, Solomon T, Braude P, Strawbridge R, Carter B. Long-term effects of COVID-19 on mental health: A systematic review. Journal of Affective Disorders. 2022;299: 118–125. doi:10.1016/j.jad.2021.11.031

Hawes MT, Szenczy AK, Klein DN, Hajcak G, Nelson BD. Increases in depression and anxiety symptoms in adolescents and young adults during the COVID-19 pandemic. Psychological Medicine. 2021; 1–9. doi:10.1017/S0033291720005358

Robinson E, Sutin AR, Daly M, Jones A. A systematic review and meta-analysis of longitudinal cohort studies comparing mental health before versus during the COVID-19 pandemic in 2020. Journal of Affective Disorders. 2022;296: 567–576. doi:10.1016/j.jad.2021.09.098

Shanahan L, Steinhoff A, Bechtiger L, Murray AL, Nivette A, Hepp U, et al. Emotional distress in young adults during the COVID-19 pandemic: evidence of risk and resilience from a longitudinal cohort study. Psychological Medicine. 2022;52: 824–833. doi:10.1017/S003329172000241X

Viner R, Russell S, Saulle R, Croker H, Stansfield C, Packer J, et al. School closures during social lockdown and mental health, health behaviors, and well-being among children and Adolescents during the first COVID-19 wave. JAMA Pediatrics. 2022;176: 400. doi:10.1001/jamapediatrics.2021.5840

2. The authors tested two theoretical modes (i.e., general-stressor model and diathesis-stressor model). However, they did not introduce the two models in the Introduction. They only briefly mention the general concepts of the two models. However, given that the study’s main focus is to examine and compare the two models, the authors should elaborate the information and descriptions of the two models. It would be much better if the authors also use the figures to explain the two models in the Introduction.

We have now expanded upon the two models in the introduction to give each theoretical model more space to be explained (pp. 3-4). 

Researchers and public health specialists raised concerns that this time of acute stress would bring about a global spike in mental illness [12]. In doing so, they implicitly argue in favor of a general-stressor model, pointing out that the many disruptions and challenges presented by the COVID-19 pandemic have led to greater levels of cumulative stress across the population (Fig 1a). This population-level stress would transdiagnostically increase levels of mental illness on a population level [13]. Indeed, infection with COVID-19 has indeed been found to negatively impact mental health [14,15] and pandemic-related stress has been found to be associated with elevated symptom severity [16,17]. However, recent large-scale studies have found less support for the general stressor model on a population level, even at the beginning of the pandemic, a period of great stress and uncertainty. [6,18,19] Indeed, populations tend to be quite resilient in the face of disaster-related stress (for review, see [20]). For example, in 2012, Hurricane Sandy-related stress only predicted elevated symptoms among children predisposed to symptom-relevant affect; those high in temperamental sadness in a prior assessment showed elevated levels of depressive symptoms while those high in temperamental fearfulness showed elevated levels of anxiety symptoms [21]. Similarly, the COVID-19 pandemic may serve as a trigger for those with vulnerabilities to specific disorders as opposed to as a population-level stressor [22].

Furthermore, the theorized relationship between pandemic-related distress and psychopathology may follow an opposite causal direction as that presumed in a general-stressor model. Loneliness caused by self-isolation may lead to greater levels of depression as hypothesized [10,23]. However, in a diathesis-stress model (See Fig 1b), the opposite causal direction is also possible [13]. In such a case, individuals with more severe depression are themselves more sensitive to distress [24]. Those with more severe baseline symptom severity may be more sensitive to the loneliness experienced during self-isolation, particularly at the beginning [18,25]. Thus, in such a scenario, the observed association between depression and pandemic-related loneliness would still exist. [16] However, it would not be because pandemic-related loneliness led to greater levels of depression. Rather, in such a model, those who entered the pandemic with higher levels of depression would feel greater levels of loneliness during lockdown.

We have also included additional citations to support each approach and explicitly traced how each part of the model would relate to each other. Additionally, we have taken the reviewer’s suggestion to include a figure that displays these relationships, as preparation for the analyses that will be performed below (Figures 1a-1b).

3. The majority of the participants were recruited from the US and the UK. Also, the authors described more COVID-19 information for the two countries. Therefore, I think that it is necessarily to do a sensitivity analysis on the US and UK samples (i.e., removing the Canada and Ireland participants to examined the tested models).

We have repeated analyses using only samples from the US and UK and the resultant findings were remarkably similar. Average scores on the metrics of interest closely overlapped. For example, in the full sample, participants’ levels of pandemic-related anxiety were M = 57.40, SD = 28.00 while in the only UK-US sample, it was M = 57.54, SD = 27.51. Effect sizes from the main analyses were extremely similar as well. In the full sample, for example, depression from T1 to T2 changed with a Cohen’s d of -.02 [-0.10; 0.06]. In the UK-US sample, Cohen’s d for this effect was -.02 [-0.11; 0.06]. We thus feel that the main findings were not driven by the inclusion of participants from Ireland and Canada. We now include the full re-analysis as a supplemental material (S1 File) for others who may have a similar concern.

4. The description of DASS is unclear. From the description of “seven-item subscales”, I know that the authors used DASS-21 instead of DASS-42. However, the authors did not make it clear that they have used DASS-21. Moreover, the citation credit should give to Lovibond and Lovibond (1995), as they are the original developers. It is fine to cite Henry and Crawford. However, Lovibond and Lovibond cannot be excluded in the citations.

We have updated the description to clarify the use of the DASS-21 and included the Lovibond & Lovibond citation as well.

5. The authors should provide scoring information for all the instruments. Also, the meaning of the directions in each instrument should be provided. Without such information, one cannot interpret the scores for the instruments.

We have provided information for each scale’s item scoring system, potential range, and meaning of higher scores.

6. From the Procedure section, one can understand that the attrition rate of the longitudinal study was about 40%. This is fine. However, the authors should provide information regarding whether the retained participants and the lost-to-follow-up participants share similar demographics. This can check if the missing is at random.

We have reviewed attrition rates and demographic information about the participants and found that those who dropped out were higher in depression and anxiety to a small degree. These differences in depression (M (SD) = 5.85 (5.64) vs 7.76 (5.97)) and anxiety (M (SD) = 3.25 (3.91) vs 4.33 (4.25)), however, were not judged to be substantial enough to establish a clinically significant difference between the groups. Furthermore, because participants were only included if they completed all three assessments, we were not concerned about this difference impacting sample scores across timepoints.

We do feel, however, that readers would want to be informed of this difference in order to better evaluate the results. As such, we have updated the manuscript in a number of ways. First, we now report these differences in the text of the Procedure section (p. 11):

Participants who returned for the final assessment showed small differences in symptom severity and moderate differences in age from those who did not (S1 File). Specifically, participants who returned were older than those who did not (M = 41.90, SD = 13.07 vs M = 34.07 SD = 10.28, d = .65, p < .001), slightly less depressed (M = 5.85, SD = 5.64 vs M = 7.76, SD = 5.97, d = .33, p = .004), and slightly less anxious (M = 3,25, SD = 3.91 vs M = 4.33, SD = 4.25, d = .27, p = .019). In order to minimize their effects on difference scores between timepoints, only participants who completed all three time-points were included in analyses.

Second, we include a table of differences in scores as a function of attrition in supplementary table S2 Table (p. 32).

Finally, we note this issue at the end of the Discussion (pp. 22-23):

Additionally, the differences between those who returned for the final assessment and those who did not is worthy of acknowledgement. Although the differences in symptom severity were small and of low clinical significance before the pandemic, it is possible that the participants who did not return for the final assessment were more at risk for symptom increase than those who did. In such a case, it is possible that had they been included, equivalence across timepoints may not have occurred. This hypothetical circumstance, however, is also inconsistent with the general stressor model that predicts symptom increase across the population. Furthermore, the current study is consistent with other large-scale studies that have also not found that symptom severity increased precipitously during the pandemic. [6,18,19] Further research may benefit from focusing on how the pandemic impacted well-being among participants at different levels of symptom severity.

7. From the statement, “Participants were recruited via the Prolific Academic Platform as part of an ongoing study on reinforcement sensitivity, emotion regulation, and affective psychopathology”, I wonder whether the authors have some publications already published to make a citation here.

We have now amended the text to read (p. 10) “Participants were recruited via the Prolific Academic Platform as part of an ongoing study on reinforcement sensitivity, emotion regulation, and affective psychopathology [35].”. 

8. Tables 2 and 3 are out of the size and cannot be read.

We have adjusted the page orientation for these tables in order to include the full tables within the margins. 

9. For all the tables, the authors should use footnotes to provide definition of T1, T2, and T3.

Tables and figures were updated to include definitions for each time-point (e.g., “T1 – Data collection at Time 1, October 17, 2018. T2 – Data collection at Time 2, April 15-22, 2019. T3 – Data collection at Time 3, April 15-20, 2020.”)_

---

## [Decision Letter · Decision Letter 2]

28 Jul 2022

PONE-D-21-18866R2Mood Symptoms Predict COVID-19 Pandemic Distress but not Vice Versa: An 18-Month Longitudinal StudyPLOS ONE

Dear Dr. Katz,

Thank you for submitting your manuscript to PLOS ONE. After careful consideration, we feel that it has merit but does not fully meet PLOS ONE’s publication criteria as it currently stands. Therefore, we invite you to submit a revised version of the manuscript that addresses the points raised during the review process. Your paper has been re-assessed by our referees. Overall, they remark the good progress made in your previous round of revisions. However, Reviewer 1 raised a set of minor comments that must be addressed by you (see attachments) before considering the manuscript as acceptable for publication in PLOS ONE. As these changes are not substantial (bit still need to be addressed with all the rigor possible), I will be pleased to evaluate them myself, instead of starting a new round of reviews. This might help to expedite an editorial decision on the manuscript. 

We look forward to receiving your revised manuscript.

Kind regards,

Sergio A. Useche, Ph.D.

Academic Editor

PLOS ONE

Journal Requirements:

Reviewers' comments:

Reviewer's Responses to Questions

**Comments to the Author**

1. If the authors have adequately addressed your comments raised in a previous round of review and you feel that this manuscript is now acceptable for publication, you may indicate that here to bypass the “Comments to the Author” section, enter your conflict of interest statement in the “Confidential to Editor” section, and submit your "Accept" recommendation.

Reviewer #1: (No Response)

Reviewer #2: All comments have been addressed

2. Is the manuscript technically sound, and do the data support the conclusions?

Reviewer #1: Yes

Reviewer #2: Yes

3. Has the statistical analysis been performed appropriately and rigorously? 

Reviewer #1: I Don't Know

Reviewer #2: Yes

4. Have the authors made all data underlying the findings in their manuscript fully available?

Reviewer #1: Yes

Reviewer #2: Yes

5. Is the manuscript presented in an intelligible fashion and written in standard English?

Reviewer #1: Yes

Reviewer #2: Yes

6. Review Comments to the Author

Reviewer #1: Mood Symptoms Predict COVID-19 Pandemic Distress but not Vice Versa:

An 18-Month Longitudinal Study

Benjamin A. Katz,1* Iftah Yovel1

KSB Review (7-27-22)

1st review comments (1-16-22): Yellow

2nd review comments (7-27-22): Blue

p. 3

“The high levels of health and financial uncertainly salient to the pandemic are strongly linked to internalizing pathology [3].”

What about “externalizing pathology”. Seems important to reference as well.

My question re: externalizing pathology was not addressed by the authors. I would assume there are a lot of data demonstrating that the pandemic was correlated with increased marital/family conflict, increased drug use, increased aggression, etc. Even if the authors chose not to explore this in their article, I think it would be important to mention, briefly, as it reflects an oft discussed problem associated with social isolation.

p. 4

“Thus, in order to assess the COVID-19 pandemic’s negative impact on mental health, it is necessary to identify participants with pre-pandemic baseline data available, and to separately assess clinical symptoms, environmental stressors, and subjective distress.”

Thoughtful, complex design.

…

“The pandemic assessment period occurred between April 15, 2020 and April 20, 2020,..”

This research is limited by the lack of measurements 6 or 12 or 18 months into the pandemic, that reflects the effects of long-term, cumulative stress.

This was addressed by the authors. Perhaps I missed it the first time, or perhaps they added it, but I appreciate their recognizing the importance of longer-term followup.

p. 5-6

Where is the demographic data. It appears pre-disaster trauma history (e.g. ACES study) was not included. Given that interpersonal trauma could be a predictor of post-disaster/pandemic reactivity, this should at minimum be mentioned as a study limitation.

This has not been mentioned/addressed by the authors, and I think should at minimum be included in their discussion of the limitations of this study. On page 17, they wrote: “Thus, those most at risk for distress during the pandemic were those experiencing both environmental stressors relevant to the distress (e.g., economic stress) as well as a history of mental illness (e.g., anxiety).”

That is good as far as it goes, using baseline data, but it makes no reference to how trauma history might predict greater distress and psychopathology as response to the pandemic. I would still want this referenced, however briefly, around page 17.

p. 6

“They represented a wide range of ages (M = 42.87, SD = 13.09, range = 19 – 75)”

It would be helpful to see the data on how many people in each age grouping, example 19-29, 30-39, etc. to see if age cohorts mattered or not.

The authors wrote: “Further demographic data is available in S1.”

The authors did not explore whether there are any cohort differences wrt outcome. They also didn’t explore whether there were significant outcome differences between white vs non-white, North America vs UK/Ireland. Again, I would like this at minimum referenced as a limitation worthy of further investigation.

p. 9-10

“The relationships between the environmental stressors and symptom severity were assessed using polychoric correlations. All other relationships (e.g., between subjective COVID-19 stress and symptom severity) were calculated using Pearson’s correlations.”

I am not well-versed in statistics, which perhaps is why it would be helpful for the reader to understand why two different measurements of correlation were chosen, and what they each differentially show and don’t show.

The authors now wrote: “The models included binary data (i.e., environmental stressors) and were therefore calculated using polychoric correlations and a diagonal weighted least squares (DWLS) estimator [27].”

As far as I can tell, this addressed my concern as stated above.

…

“After observing the unexpectedly small effect sizes of change between time-points, we next estimated whether these effects were statistically equivalent to zero. This was done using the two one-sided tests (TOST) procedure for equivalence testing [26].”

It is my understanding the more tests you run, the more you’re likely to “find something” of statistical significance. Is this taken into account? If so, specify how.

I’m not sure this was addressed by the authors, but I am not a statistician so perhaps it was considered when the authors wrote: “Models were assessed using robust fit statistics using the recommended cutoffs [28,29] of: non- significant chi-square test, CFI > .95, RMSEA < .06, SRMR < .08.”

p. 13

“Distress intolerance, on the other hand, was not found to be significantly equivalent (p = .096) across timepoints. As such, distress intolerance was judged to be different between T1 and T2, to a significant albeit trivial degree.”

Is p= .096 considered significant?? Shouldn’t it be lower than .05? Perhaps this should be explained as to why this is statistically significant, however “trivial”.

The authors have not addressed my concern at all. How can something be “not significantly equivalent” and “different... to a significant albeit trivial degree.” I understand p= .096 fits with “not significantly equivalent”, but then how do the authors conclude T1 and T2 are “different to a significant albeit trivial degree”? “Not equivalent” is not the same as “different to a significant degree”. Where is the data to support that second statement? Am I missing something?

p. 17

“While unexpected, these findings were consistent with patterns observed following other disasters. Populations appear to be resilient to large-scale environmental stressors [9].”

How much is this a function of this being very early in the pandemic? Would it change after 1 year? After getting the vaccine, and then new variants causing a resurgence?

Since the authors did followup at 12 and 18 months, this concern was addressed.

Also, the subject demographics does not assessment of interpersonal trauma history, e.g, along the lines of the ACES studies. In my clinical experience, the trauma history pre-“disaster” has a significant effect on post-disaster coping. At the minimum, the fact that interpersonal trauma history was NOT assessed should be mentioned.

As I highlighted above, p. 5-6, no mention was made of pre-pandemic trauma history. In my clinical experience and that of my colleagues, there was an intersection between complex, childhood trauma and response to the pandemic. Again, I would want that stated as a limitation of this study.

…

“…those experiencing both environmental stressors relevant to the distress (e.g., economic stress) as well as a history of mental illness (e.g., anxiety).”

Again, trauma history is not included. Some of the anxiety, depression, isolation, etc. could be a consequence of post-trauma sequelae. Again, this limitation of this study needs mentioning.

Again, history of mental illness is an important variable, but trauma history may be as or more important. Mention this is worthy of further study.

p. 18

“Although it occurred during peak rates of COVID-19 fatality (see Figure 1), other events during the pandemic may have led to other distress peaks as well, such as when participants began to self-isolate [33].”

This is a good point.

..

“Identifying which participants are most vulnerable to mental illness in the face of pandemic-related stress is the first step in developing targeted interventions [35]. In doing so, the clinical science community may rise to the formidable challenge that COVID-19 has set before it.”

Agreed, but I don’t think the researchers take into account what I believe (and I assume research has shown) the importance of interpersonal, particularly severe childhood trauma history (a la the ACES research) and neglect. If you don’t take into account childhood trauma and neglect, your interventions will not be as patient-specific and effective as desired.

On page 18, the authors wrote: “Collaborative work between multiple laboratories will also offer opportunities to compare findings among diverse populations and assessment methods in order to more efficiently identify at-risk populations [34].

Not to beat a dead horse, but what I highlighted speaks directly to my concern that pre-pandemic, and especially early childhood, relational trauma history was not assessed. Since the issue of relational trauma addresses both matters of “population” and “assessment”, this would be the place to mention trauma history as important in “further/future study.”

Reviewer #2: I am happy with the revised manuscript. Apparently, the authors have well addressed all the prior comments. I have no more comments and I would like to thank the authors again for addressing all the prior issues.

7. PLOS authors have the option to publish the peer review history of their article (what does this mean?). If published, this will include your full peer review and any attached files.

Reviewer #1: No

Reviewer #2: No

---

## [Author Response · Author response to Decision Letter 2]

8 Aug 2022

Reviewer #1: 

p. 3

“The high levels of health and financial uncertainly salient to the pandemic are strongly linked to internalizing pathology [3].”

My question re: externalizing pathology was not addressed by the authors. I would assume there are a lot of data demonstrating that the pandemic was correlated with increased marital/family conflict, increased drug use, increased aggression, etc. Even if the authors chose not to explore this in their article, I think it would be important to mention, briefly, as it reflects an oft discussed problem associated with social isolation.

We have now expanded upon the role of externalizing pathology and recommended that the current findings should be considered against similar studies on externalizing disorders (p. 25):

Additionally, the current study operationalized mental health in terms of metrics of distress, loneliness, depression, and anxiety. However, other metrics of behavioral health, such as substance use and behavioral addictions, have also been associated with social isolation during the pandemic’s onset [9]. The current study may be considered against others related to externalizing disorders in order to assess trends across different types of psychopathology.

p. 5-6

Where are the demographic data. It appears pre-disaster trauma history (e.g. ACES study) was not included. Given that interpersonal trauma could be a predictor of post-disaster/pandemic reactivity, this should at minimum be mentioned as a study limitation.

This has not been mentioned/addressed by the authors, and I think should at minimum be included in their discussion of the limitations of this study. On page 17, they wrote: “Thus, those most at risk for distress during the pandemic were those experiencing both environmental stressors relevant to the distress (e.g., economic stress) as well as a history of mental illness (e.g., anxiety).”

That is good as far as it goes, using baseline data, but it makes no reference to how trauma history might predict greater distress and psychopathology as response to the pandemic. I would still want this referenced, however briefly, around page 17. 

Following the paragraph mentioned by the reviewer now on pp. 21-22, we now include the importance of adverse childhood experiences and other interpersonal factors in the discussion and explicitly point to where other longitudinal studies may be used to assess such questions (p. 23):

Indeed, it also more generally highlights the importance of longitudinal data in identifying those most at risk for elevated levels of mental illness during periods of great distress. For example, individuals with adverse childhood experiences may also be at risk for increased levels of mental illness during times of pandemic-related stress [47]. In this case, ongoing, lifetime longitudinal studies are specially equipped to assess the relationship between diatheses related to earlier experiences and later distress [16].

p. 6

“They represented a wide range of ages (M = 42.87, SD = 13.09, range = 19 – 75)”

The authors wrote: “Further demographic data is available in S1.”

The authors did not explore whether there are any cohort differences wrt outcome. They also didn’t explore whether there were significant outcome differences between white vs non-white, North America vs UK/Ireland. Again, I would like this at minimum referenced as a limitation worthy of further investigation.

Owing to the limited sample size and distribution of participants across varied locations, we are concerned that applying the number of comparisons in this suggestion may yield false positives in terms of group differences in effect size. However, consistent with the reviewer’s request, we now reference the role of demographics on COVID-related depression and anxiety in the discussion, highlighting the role of syndemic interactions between minoritized identity and the effects of the pandemic. We agree with the reviewer that interactions such as these are worthy of future inquiry, and, consistent with the reviewer’s request, note this as a limitation in the current study’s analysis and highlight this point as being worthy of further investigation on p. 23:

These findings may complement others that use demographic correlates to identify those at higher risk for elevated mental illness during the pandemic. Factors such as age, gender, race, ethnicity, location, and education are associated with internalizing pathology at the onset of the pandemic, alongside pandemic-related stressors [44]. These different risk factors interact syndemically, wherein pandemic-related stressors have greater impacts upon mental health for those with minoritized racial identities [45] or live in lower-income communities [46]. While the current study’s sample size is too small to undertake such interaction analyses, other large-scale studies will be better equipped to do so.

p. 13

“Distress intolerance, on the other hand, was not found to be significantly equivalent (p = .096) across timepoints. As such, distress intolerance was judged to be different between T1 and T2, to a significant albeit trivial degree.”

Is p= .096 considered significant?? Shouldn’t it be lower than .05? Perhaps this should be explained as to why this is statistically significant, however “trivial”.

The authors have not addressed my concern at all. How can something be “not significantly equivalent” and “different... to a significant albeit trivial degree.” I

understand p= .096 fits with “not significantly equivalent”, but then how do the authors

conclude T1 and T2 are “different to a significant albeit trivial degree”? “Not

equivalent” is not the same as “different to a significant degree”. Where is the data to

support that second statement? Am I missing something?

Response:

The TOST approach entails three steps when assessing group difference-vs-equivalence. First, we assess difference using null-hypothesis significance testing (NHST). For distress intolerance, his was the first test which was significant (i.e., “significantly different from T1 to T2 using NHST (d = .10, p = .049)”). Next, we assessed equivalence using the two one-sided tests (TOST) approach, which led to a non-significant outcome (i.e., “was not found to be significantly equivalent using TOST (p = .096)”). Finally, we assessed the size of this difference using Cohen’s d. Consistent with convention, effect sizes up to d = .10 were considered trivially small. We understand that the reviewer was also previously concerned that this would be an excess of tests for a single effect size. However, we would like to emphasize that this three-step process is the gold-standard approach for equivalence testing. We have added clarifications to the Analysis Plan section of the manuscript so that the systematic nature of these steps would be more clear (p. 12):

Assessing the extent to which depression and anxiety changed or remained the same between time points was done in three steps. First, we performed conventional null hypothesis significance tests (NHST) to examine differences between time-points. Specifically, within-group t tests were used, with symptom and trait measures (e.g., DASS-Depression) entered as the repeated measures across consecutive time-points (i.e., T1-T2 or T2-T3). In this case, a significant finding would indicate that the repeated measures changed between time-points. 

After observing the unexpectedly small effect sizes of change between time-points, we next estimated whether these effects were statistically equivalent to zero. This procedure is performed using a combination of the previous NHSTs and an additional two one-sided tests (TOST) procedure for equivalence testing [39]. In this procedure, an a priori smallest effect size of interest (SESOI) is calculated. Two one-sided t-tests then assess whether the full confidence interval of the change score estimate falls within the positive and negative SESOI (e.g., Cohen’s d = -.20 < X < .20). If so, the change score is judged as statistically equivalent to zero and scores are considered equivalent to each other. Finally, in the case of significant difference, we examined Cohen’s d of difference in order to assess the size of the difference between timepoints.

p. 17

“…those experiencing both environmental stressors relevant to the distress (e.g., economic stress) as well as a history of mental illness (e.g., anxiety).”

Again, trauma history is not included. Some of the anxiety, depression, isolation, etc. could be a consequence of post-trauma sequelae. Again, this limitation of the study needs mentioning.

Again, history of mental illness is an important variable, but trauma history may be as

or more important. Mention this is worthy of further study.

…

those experiencing both environmental stressors relevant to the distress (e.g.,

economic stress) as well as a history of mental illness (e.g., anxiety).”

Again, trauma history is not included. Some of the anxiety, depression, isolation, etc.

could be a consequence of post-trauma sequelae. Again, this limitation of this study

needs mentioning.

Again, history of mental illness is an important variable, but trauma history may be as

or more important. Mention this is worthy of further study

Response:

We have taken four steps to remedy this concern. First, we now mention in the Method section that childhood experiences were not assessed (p. 10): 

They completed a series of self-report questionnaires related to reinforcement sensitivity, emotion regulation and affective pathology followed by an unrelated behavioral task (e.g., for similar procedure, see [36]). Questionnaires only assessed recent levels of psychopathology. Histories of psychopathology or childhood risk factors (e.g., adverse childhood events) were not assessed.

Second, in the discussion on p. 23, we now consider the role of lifetime longitudinal studies in identifying those who experienced ACEs (“Indeed, it also more…and later distress [16].” See response to p. 5-6 above for full quote).

Third, in the limitations section, we now expand upon the extant discussion of ACEs to highlight the importance of including measures of participants’ past experiences that occurred prior to the first baseline assessment (p. 24-25):

Finally, the current study approaches risk factors for mental illness from the perspectives of contemporary intrapersonal processing, acute stress, and internalizing psychopathology. However, many risk factors for elevated mental illness occurred prior to the original baseline measures included in the study. Elevated levels of psychopathology prior to assessment [24] or adverse childhood events [47] are both useful for identifying those at risk for greater levels of distress while facing pandemic-related stressors. Future studies using similar designs may use retrospective assessments to capture some of these risk factors as well. 

Finally, we now highlight the importance of assessing clinical history as a future direction of research in the closing paragraph (p. 25):

Collaborative work between multiple laboratories will also offer opportunities to compare findings among diverse populations, assessment methods, and clinical histories in order to more efficiently identify at-risk populations [50].

p. 18

“Identifying which participants are most vulnerable to mental illness in the face of pandemic-related stress is the first step in developing targeted interventions [35]. In doing so, the clinical science community may rise to the formidable challenge that COVID-19 has set before it.”

Agreed, but I don’t think the researchers take into account what I believe (and I assume research has shown) the importance of interpersonal, particularly severe childhood trauma history (a la the ACES research) and neglect. If you don’t take into account childhood trauma and neglect, your interventions will not be as patient-specific and effective as desired.

On page 18, the authors wrote: “Collaborative work between multiple laboratories will

also offer opportunities to compare findings among diverse populations and assessment methods in order to more efficiently identify at-risk populations [34].

Not to beat a dead horse, but what I highlighted speaks directly to my concern that prepandemic, and especially early childhood, relational trauma history was not assessed. Since the issue of relational trauma addresses both matters of “population” and “assessment”, this would be the place to mention trauma history as important in

“further/future study.”

Please see the response to p. 17 for our four changes to the manuscript that were made in order to more explicitly highlight the role of childhood in symptom change during the pandemic.

---

## [Editor Report · Decision Letter 3]

19 Aug 2022

Mood Symptoms Predict COVID-19 Pandemic Distress but not Vice Versa: An 18-Month Longitudinal Study

PONE-D-21-18866R3

Dear Dr. Katz,

We’re pleased to inform you that your manuscript has been judged scientifically suitable for publication and will be formally accepted for publication once it meets all outstanding technical requirements.

Kind regards,

Sergio A. Useche, Ph.D.

Academic Editor

PLOS ONE

---

## [Editor Report · Acceptance letter]

25 Aug 2022

PONE-D-21-18866R3 

Mood symptoms predict COVID-19 pandemic distress but not vice versa: An 18-month longitudinal study 

Dear Dr. Katz:

I'm pleased to inform you that your manuscript has been deemed suitable for publication in PLOS ONE. Congratulations! Your manuscript is now with our production department. 

Kind regards, 

on behalf of

Dr. Sergio A. Useche 

Academic Editor

PLOS ONE